# Reoxidation of Al-Killed Steel by $Cr_2O_3$ from Tundish Cover Flux

**Feng Wang [1], Daoxu Liu [2], Wei Liu [2], Shufeng Yang [2],\* and Jingshe Li [1,2],\***

[1] Engineering Research Institute, University of Science and Technology Beijing, Beijing 100083, China; b20150439@xs.ustb.edu.cn

[2] School of Metallurgical and Ecological Engineering, University of Science and Technology Beijing, Beijing 100083, China; liudaoxuustb@163.com (D.L.); youiithe@foxmail.com (W.L.)

\* Correspondence: yangshufeng@ustb.edu.cn (S.Y.); lijingshe@ustb.edu.cn (J.L.); Tel.: +861062334277 (S.Y.)

**Abstract:** Reoxidation has long been a problem when casting ultra-low oxygen liquid steel. An experimental study of the reoxidation phenomenon caused by $Cr_2O_3$-bearing cover flux of Al-killed steel is presented here. $MgO-CaO-SiO_2-Al_2O_3-Cr_2O_3$ tundish cover flux with various $Cr_2O_3$ contents were used to study the effects of $Cr_2O_3$ on total oxygen content (T[O]) and alumina and silicone loss of Al-killed steel at 1923 K (1650 °C). It was found that $Cr_2O_3$ can be reduced by Al to cause reoxidation, and the reaction occurs mainly within 2 to 3 min after the addition of the tundish cover flux with 5% and 10% $Cr_2O_3$ concentration. T[O] and Al loss increase with higher $Cr_2O_3$ concentration flux. Two controlled experiments were also made to investigate the oxygen transported to the steel by the decomposition of $Cr_2O_3$. It was calculated that when Al is present in steel, more than 90% of the reoxidation of $Cr_2O_3$ is caused by Al, and the rest is caused by decomposition.

**Keywords:** reoxidation; tundish cover flux; $Cr_2O_3$; decomposition reaction

## 1. Introduction

The total oxygen content (T[O]) of high-grade steel has been required to be as low as possible in recent years [1,2]. For example, line pipe steel requires sulfur, phosphorus, and T[O] all to be less than 30 ppm, and bearing steel requires T[O] to be less than 10 ppm [3]. This low oxygen liquid steel is very hard to cast due to reoxidation during the process. The tundish used in continuous casting is the last vessel in contact with molten steel during steel production, with this final step being the most important in protecting the steel from oxygen and maintaining cleanliness [4,5]. However, due to the extremely low oxygen content in liquid steel, it can be easily reoxidized by the oxygen from the refractory, the slag, and the air environment [6,7]. The reoxidation phenomenon has been a problem for tundishes and is the obstacle to producing clean steel [8–10].

The role of tundish cover flux is to protect the molten steel from the air. However, in recent years, reoxidation of steel by the cover flux has been observed and widely investigated. Researchers have found that the reoxidation capacity of the slag was related to the oxygen potential in the slag, such as the FeO, $SiO_2$, and MnO content in the slag phase [11–13]. The high oxygen potential slag easily reacts with deoxidizing agents such as the Al and Ti contained in liquid steel. Some researchers [14] studied the effect of refining slag components on the reoxidation of molten steel through the double-membrane theory and found that reducing the $CaO/Al_2O_3$ ratio and FeO content in the slag can reduce the reoxidation of steel by refined slag.

Recently, $Cr_2O_3$ has been found in tundish cover flux [15]. Cr is usually used as an alloying element for stainless steel, working as a protecting element that generates $Cr_2O_3$ on the metal surface to prevent further oxidation [16]. Due to the steel-slag equilibrium reaction, some $Cr_2O_3$ shows up

in the refining slag of stainless steel [17,18]. The reduction of $Cr_2O_3$ during the refining process will increase T[O] and thus will do harm to steel cleanliness [19]. It has been reported that the amount of $Cr_2O_3$ forming in slag could be reduced by lowering the basicity of the slag or by adding $Al_2O_3$, the reason for this being the increase in the liquid phase fraction of the slag [19].

When using cover flux containing $Cr_2O_3$, $Cr_2O_3$ may work as a medium carrying oxygen from the air that reoxidizes the liquid steel instead of protecting it. Cover flux is a semi-molten slag that, when compared to other totally melted refining slags, makes the reduction of $Cr_2O_3$ in the flux special and worth studying. Reoxidation is related to the Si and Al content in steel and the slag composition. Therefore, in the present work, reoxidation of steel, with and without Al, by $Cr_2O_3$ in the tundish cover flux was investigated using experimental methods. The effects of the $Cr_2O_3$ content of flux on the T[O] and the loss of Al and Si in steel was investigated, and the different results of reoxidation by reduction and decomposition were compared and discussed.

## 2. Experimental Procedures

A tube resistance furnace (Baotou Yunjie Furnace Ltd., Baotou, China) was used in the experiments to melt the steel and the tundish cover flux. The schematic diagram of the tube resistance furnace is shown in Figure 1. Four hundred grams of pure industrial iron (the composition is shown in Table 1) was put into a magnesia crucible (inner diameter: 42 mm, outside diameter: 51 mm, height: 101 mm, Boshan Refratory Material Ltd., Zibo, China) and then melted in an argon gas atmosphere at 1873 K (1600 °C). Al was added to the molten steel at a concentration of 0.2 mass% to make Al-killed steel. Then, after adjusting the molten steel temperature to 1923 K (1650 °C) and holding for 30 min, 20 g tundish cover flux of different $Cr_2O_3$ concentrations were added wrapped with iron foil. The crucible was held at 1923 K (1650 °C) for another 30 min to ensure a uniform covering of the molten steel by slag. The composition of the cover flux is presented in Table 2.

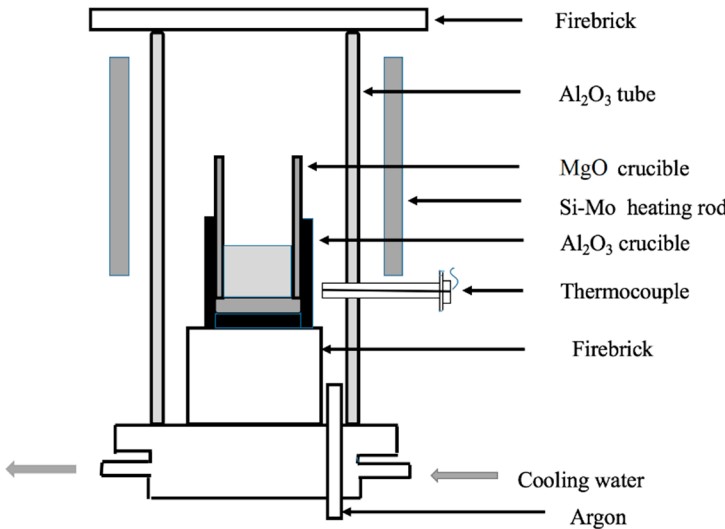

**Figure 1.** Schematic diagram of experimental apparatus.

**Table 1.** Chemical composition of YT01 (wt. %).

| C | Si | Mn | P | S | Al | Ni | Ti | N | Ca | Fe |
|---|---|---|---|---|---|---|---|---|---|---|
| 0.0016 | 0.0033 | 0.01 | 0.0053 | 0.0017 | 0.003 | 0.0038 | 0.001 | 0.002 | - | 99.95 |

**Table 2.** Compositions of slags (wt. %).

| Experiment | CaO | $Al_2O_3$ | MgO | $SiO_2$ | $Cr_2O_3$ |
|:----------:|:---:|:---------:|:---:|:-------:|:---------:|
| 1 | 50 | 30 | 10 | 10 | 0 |
| 2 | 48 | 27 | 10 | 10 | 5 |
| 3 | 46 | 24 | 10 | 10 | 10 |
| 4 | 48 | 27 | 10 | 10 | 5 |
| 5 | 46 | 24 | 10 | 10 | 10 |

During the experiments, steel samples were taken as follows: (a) before adding the tundish cover flux, and (b) 5, 10, and 15 min after adding the tundish cover flux. Tundish cover flux samples were collected after the experiments.

Oxygen content was determined using the infrared method. The calculated aluminum, silicon, and chromium contents of the steel were determined using ICP-AES (NCS plasma 2000, NCS Testing Technology Co., Ltd., Beijing, China). FeO content in the tundish cover flux was analyzed using XRF (ZSX Primus II, Rigaku Corporation, Tokyo, Japan).

## 3. Results and Discussion

### 3.1. Reoxidation of Al-Killed Steel by $Cr_2O_3$

The changes in Al and T[O] content in Experiments 1, 2, and 3 are shown in Figures 2 and 3. It can be seen that the Al content in the steel dropped sharply and the O content rose sharply when flux with $Cr_2O_3$ was added. This indicated that $Cr_2O_3$ in the tundish cover flux can be reduced by Al in the molten steel and lead to the reoxidation. The changes in Al and O contents mainly occurred in the first 5 min after the addition of the tundish cover flux. The reduction of $Cr_2O_3$ by Al occurred immediately after the addition of the tundish cover flux, and it was very quick. The increase in $Cr_2O_3$ from 5% to 10% showed no obvious difference on the Al and O contents. This indicates that for the slag-steel system used in this study, as the content of $Cr_2O_3$ increased, the transport of O from slag to steel showed no increase, and that the 5% $Cr_2O_3$ flux had already reached the limitation. After 5 min, T[O] began to drop sharply, suggesting that the reduction reaction was very fast and reached the equilibrium quickly. Thus, after the first 5 min of the reaction, $Al_2O_3$ inclusions floated up and were removed, resulting in the drop in T[O] [11]. However, compared with no $Cr_2O_3$ flux, the T[O] at 15 min was much higher in the other two experiments, indicating the reoxidation of steel by the $Cr_2O_3$ flux.

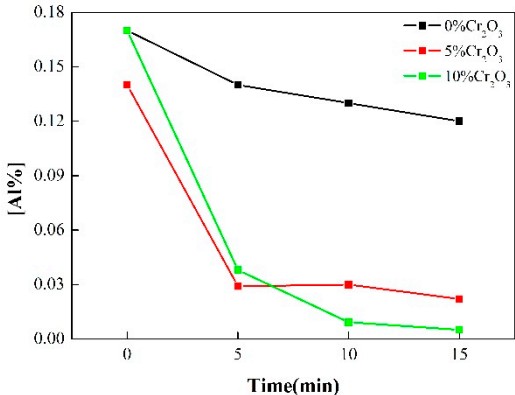

**Figure 2.** Variations in Al concentrations of steel with time with different concentrations of $Cr_2O_3$ cover flux added.

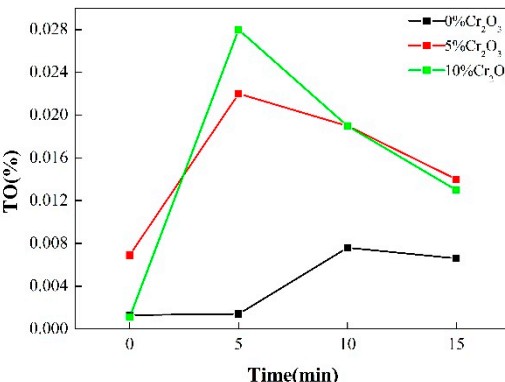

**Figure 3.** Variations in total oxygen content (T[O]) of steel with different concentrations of $Cr_2O_3$ cover flux added.

In order to determine the speed of the reduction reaction, another experiment with more frequent sampling was designed. The sampling time was 1, 2, 3, 4, and 5 min after the tundish cover flux was added, as shown in Figure 4. It can be seen that the reaction of Al reducing $Cr_2O_3$ occurred mainly within 2 min after adding the cover flux for the 5% $Cr_2O_3$ flux and within 3 min for the 10% flux. Moreover, the reaction rates were almost the same in the first minute after adding the flux, and the Cr content rose to 0.08% in the first minute for both. Between the first and second minute, the rate remained at 0.08%/min for the 5% $Cr_2O_3$ flux, while the 10% flux reached the highest rate of 0.16%/min. Then at 3 min, the reduction of $Cr_2O_3$ stopped for the 5% $Cr_2O_3$ flux, while the reduction of the 10% flux lasted another minute at a rate of 0.08%/min. The reduction reaction in this study happened at a very high rate and required little time. The increase of the equilibrium of Cr in steel was caused by the change in $Cr_2O_3$ content of the flux. The highest reaction rate occurred with the higher $Cr_2O_3$ concentration flux, and this may be explained by the doubling of the concentration of $Cr_2O_3$ in the slag being the driving force behind the transport of $Cr_2O_3$ from the slag to the interface [19].

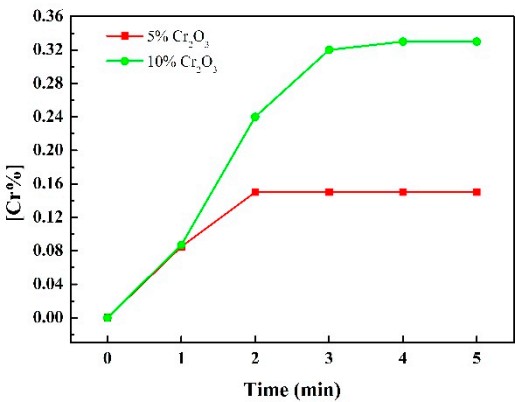

**Figure 4.** Variations in Cr concentrations of steel with time for Experiments 2 and 3.

The changes in Si and Cr in Experiments 1, 2, and 3 are shown in Figures 5 and 6. It can be seen from Figure 5 that before the addition of flux with $Cr_2O_3$, the Si content in the steel increased first and then decreased, and after the flux was added, the Si content in the steel was significantly reduced. Furthermore, when adding flux with 10% $Cr_2O_3$, the Si content in the steel first decreased and then increased. It is known that Al in steel preferentially reacts with $Cr_2O_3$ in the cover flux. This means $Cr_2O_3$ is superior to $SiO_2$ when reacting with Al in steel. Thus, it is assumed that Reaction (1) will be weakened by Reaction (2) [20,21] because of the limited oxygen supplement [17,18]. The dashed line in Figure 6 represents the ideal value of Cr content that can be reduced in the flux according to a mass balance calculation. It was found that almost all of the $Cr_2O_3$ in the cover flux was reduced,

which made Cr close to the ideal value. When increasing the $Cr_2O_3$ content from 5% to 10%, there was even less Si due to the oxidation of the Si in the steel. This indicates that if not enough oxygen is provided by the flux, the reoxidation of molten steel by $Cr_2O_3$ will first involve Al, while the reaction will involve both Al and Si when sufficient oxygen is supplied by the flux.

$$4[Al] + 3(SiO_2) \rightarrow 3[Si] + 2(Al_2O_3) \tag{1}$$

$$[Al] + (Cr_2O_3) \rightarrow [Cr] + (Al_2O_3) \tag{2}$$

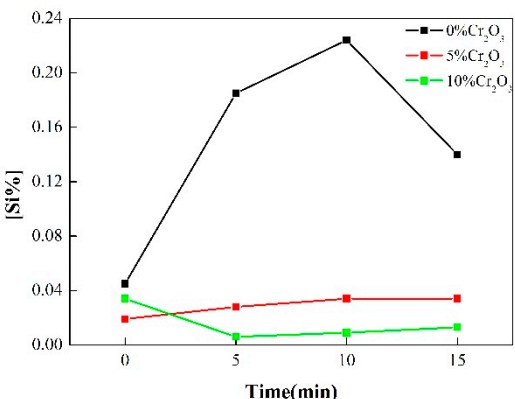

**Figure 5.** Variations in Si concentrations of steel with time for Experiments 1, 2, and 3.

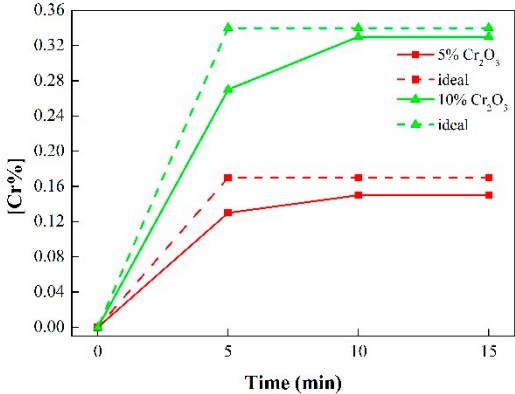

**Figure 6.** Variations in Cr concentrations of steel with time for Experiments 2 and 3.

### 3.2. Reoxidation of Non-Al Steel by $Cr_2O_3$

Compared to Experiments 2 and 3, no Al was added to the molten steel in Experiments 4 and 5. The changes in the Cr and Si contents in Experiments 4 and 5 are shown in Figures 7 and 8, respectively. It can be seen from these figures that even if Al is not added to the steel, the Cr content in the steel will still rise dominantly. In Experiment 4, the Si content of the steel first increased and then decreased, which indicates that $Cr_2O_3$ and $SiO_2$ in the tundish cover flux can transport oxygen to the steel by decomposition [22]. By comparing the magnitude of the rise, it can be seen that the decomposition of $Cr_2O_3$ dominates, and that the Si content in the steel begins to decrease due to the increase in the oxygen potential of the steel caused by the continued decomposition of $Cr_2O_3$, which causes the Si in the steel to be oxidized. The Si content in Experiment 5 first decreased and then remained unchanged. This was due to the $Cr_2O_3$ content in the tundish cover flux being higher, causing the decomposition of $Cr_2O_3$ to be very intense from the beginning. Therefore, the oxidation rate of the Si in the steel was greater than the decomposition rate of $SiO_2$ in the tundish cover flux, which led to a decrease in the Si content of the steel. The subsequent decomposition of the $Cr_2O_3$ further reduced the Si content in the steel until it reached a very low value, after which the Cr in the steel began to be oxidized (as shown

in Figure 8). Combining the change in Al content of the Experiment 3 steel and the stoichiometry of Reaction (2), it can be calculated that when Al is present in steel, the contribution of Al reduction to the increase in Cr content of steel is more than 90%, and that of $Cr_2O_3$ decomposition is less than 10%. Figures 9 and 10 compare the T[O] changes with time between Experiments 2 and 4 and Experiments 3 and 5, respectively. It can be seen from Figures 9 and 10 that the oxygen content in Experiments 4 and 5 was significantly greater than that in Experiments 2 and 3. The main reason for this was that Al was not added, and thus the deoxidation element was mainly Si in Experiments 4 and 5.

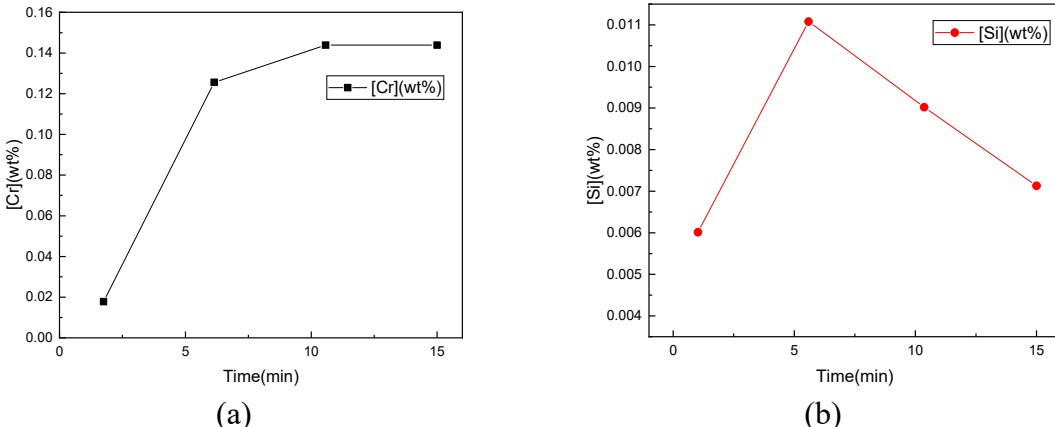

**Figure 7.** Variations in Cr and Si concentrations of steel with time for Experiment 4: (**a**) Cr, (**b**) Si.

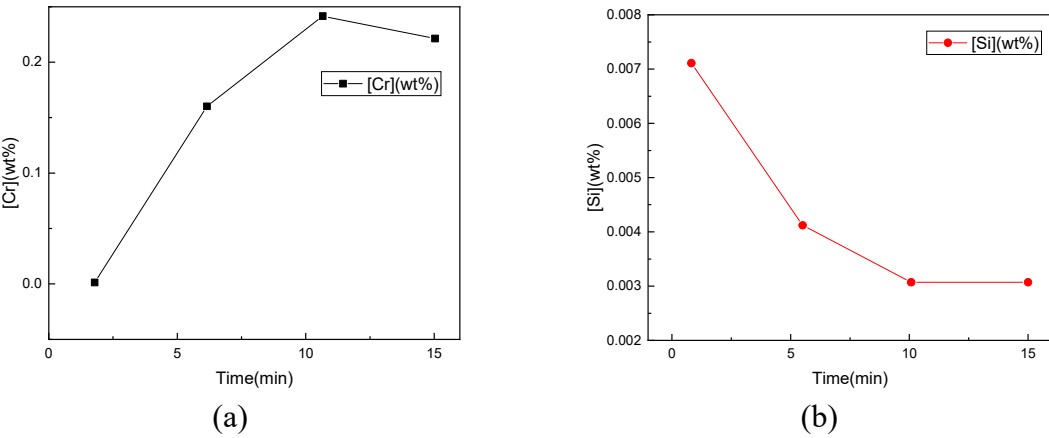

**Figure 8.** Variations in Cr and Si concentrations of steel with time for Experiment 5: (**a**) [Cr], (**b**) [Si].

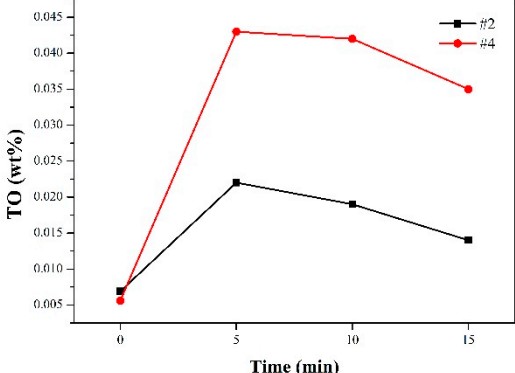

**Figure 9.** Variations in T[O] of steel with time for Experiments 2 and 4.

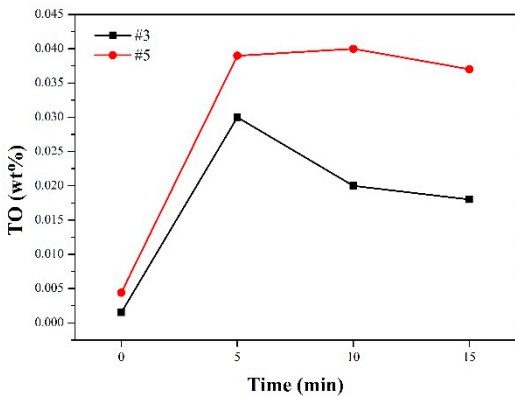

**Figure 10.** Variation in T[O] of steel with time for Experiments 3 and 5.

## 4. Conclusions

In this work, the unwanted reoxidation of steel by $Cr_2O_3$-bearing tundish cover flux was investigated through high temperature experiments. It was found that $Cr_2O_3$ can be reduced by Al to cause the reoxidation of steel. Moreover, as the $Cr_2O_3$ concentration increased, the reoxidation of the steel became more and more serious, and the Al loss also increased. It was also found that the reduction of $Cr_2O_3$ by Al mainly occurred within 2 to 3 min after the addition of the tundish cover flux with $Cr_2O_3$ concentrations of 5% and 10%. During the reduction of $Cr_2O_3$ by Al, the reaction rate reached a maximum value 2 min after the addition of the tundish cover flux with 10% $Cr_2O_3$, which may be related to kinetic conditions, and requires further investigation. Moreover, the reoxidation of molten steel by $Cr_2O_3$ caused the oxygen potential of the molten steel to rise, thereby suppressing the reoxidation of $SiO_2$. It was concluded from two controlled experiments that $Cr_2O_3$ can also transport oxygen to the steel through decomposition. Furthermore, it was calculated that when Al is present in steel, more than 90% the reoxidation of $Cr_2O_3$ is caused by Al, and the rest is caused by decomposition.

**Author Contributions:** S.Y. and F.W. conceived and designed the experiments and interpreted the data; D.L. and W.L. wrote the paper; F.W. and J.L. analyzed the data and collected the literature; D.L. and F.W. performed the experiments.

**Funding:** This research was funded by National Natural Science Foundation of China, grant number 51734003 and 51674023.

**Conflicts of Interest:** The authors declare no conflict of interest.

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
