# Peer review of "Reoxidation of Al-Killed Steel by Cr2O3 from Tundish Cover Flux"

_metals, doi:10.3390/met9050554_

Reviewer 1 Report

Please see attached

Author Response

Please see the attached word file of response.

Reviewer 2 Report

Dear Authors,

The effect of Cr2O3 on the reoxidation of Al-killed steel was experimentally studied in the paper. The experimental tests were performed with different composition of slags. The Authors investigate also the influence of adding aluminum to the molten steel on the variation of total oxygen content and chemical composition of the steel. The influence of Aluminium and Cr2O3 is discussed in the paper. It was observed that the presence of Cr2O3 increase total oxygen content in steel and decrease Aluminum content.

Main remarks:

1.   In my opinion, the introduction should be extended by:

- discussion concerning the influence of oxygen in different steels,

- more comprehensive literature review - description and comparison of the method of deoxidation as well as reoxidation of steel by other chemical compounds (in example [1]),

- The influence of Cr2o3 was investigated and discussed in [2]. In this paper, it was observed that „from the viewpoint of minimizing inclusions through Al reoxidation in steel, the presence of Cr203 in the slag is thermodynamically harmful even though its content is small”. This paper should be cited and Authors should refer to the results and conclusions presented in [2, 3],

- It should be also clearly stated by the Authors what new is presented in the proposed paper.

2.   Line 99: How theoretical Cr content was calculated? It should be clarified in the paper.

3. The English language and style should be carefully corrected. In some cases, there are words which can have inappropriate meanings. There are phrases which are to long or grammatically incorrect which makes difficult to read and understand.

4.  There are many conclusions in the paper which are not thoroughly explained (Line 70, 96, 99, 123, 131). More detailed discussion would facilitate understanding of presented phenomena.

Other remarks

1.   Line 38 – „Recently, Cr2O3 has been found in tundish cover flux.” – reference should be given here.

2.   Line 45 – abbreviations (ID, OD,H) should be explained

3.   Descriptions of chemical compounds should be corrected (line 36, 73, 152)

4.   Figure 1. I am not sure about the correctness of „Stent”. It should be carefully checked.

5.   Table  1. The sum of chemical composition doesn’t give 100%.

6.   Line 69:  „The effect of flux with 5% and 10% Cr2O3 shows no obvious difference on the Al and O contents.” Maybe -  The increase of Cr2O3 from 5% to 10% .....

7.   Line 71: „This indicates that as the 69 content of Cr2O3 increases, the transport of O from slag to steel shows no increase and the 5% one has 70 reached the limitation.” Is this assumption was verified for steels with a different chemical composition?

8.   Line 80: „To determine how quick the reduction reaction is,” maybe „In order to determine the reaction rate…”

9.   Line 81: „The sampling time is 1, 2, 3, 4 and 5 min…”. Past tense should be used.  

10.  Lines 84-90 – It is difficult to read, grammar should be corrected

11.  Line 102: I proposed to change the word „depressed”, maybe „weakened”?

12.  Line 94: I propose to add „in steel” after Si and Cr content.

13.  Lines 105-107: It can be understood that the reoxidation of molten steel is desirable.

14.  Line 115: „Experiments 4 and 5 differed from Experiments 2 and 3 in that no Al was added to the molten steel.” – grammar should be corrected.

15.  Line 116: I propose to use „respectively” at the end of this phrase.

16.  Lines 132-134- I propose to divide this phrase into two sentences

17.  Figure 7: I propose to give both diagrams next to each other instead of inside.

18.  Figure 8:  as above

19.  Line 144. If the reoxidation of Cr2O3 in tundish cover flux on Al-killed steel is a negative phenomenon, then it would be beneficial to emphasize this in this sentence.

[1] Fang Jiang, Yan Liu,You Xie, Guoguang Cheng. Reoxidation of Al‐Killed Steel by Ca(OH)2 in the High Basicity Slag. steel research int. 83 (2012) No. 9. https://doi.org/10.1002/srin.201200057

[2] Sung-Koo Jo and Seon-Hyo Kim. Thermodynamic assessment of CaO-Si02-AI203-MgO-Cr203-MnO-FetO slags for refining chromium-containing steels. Steel Research 71 (2000) NO.8

[3] (no. 11 in the paper) F. Jiang; G. Cheng; Y. Xie; G. Qian; Q. Rui; Y. Song. Reoxidation of Al-killed molten steel by Fe2O3 and 183 Cr2O3 in the magnesia-chromite refractory. Steel Res. Int. 2013,84, 1098-1103.

Best Regards,

Author Response

Please see the attached response file.

Round  2

Reviewer 1 Report

please see attached

Reviewer 2 Report

Dear Authors,

The effect of Cr2O3 on the reoxidation of Al-killed steel was experimentally studied in the paper. The experimental tests were performed with different composition of slags. The Authors investigate also the influence of adding aluminum to the molten steel on the variation of total oxygen content and chemical composition of the steel. The influence of Aluminium and Cr2O3 is discussed in the paper. It was observed that the presence of Cr2O3 increase total oxygen content in steel and decrease Aluminum content.

The Authors responded to all remarks and comments and they introduced changes in the manuscript. However, in my opinion one task is still unclear. This is related with that the aim of the study is not clearly justified. In the lines 48-49 it is written ” An interesting problem that whether the Cr2O3 in flux protects steel from oxidizing or works as an oxygen source to re-oxidize steel remains unclear.”, however in the Ref [19] this problem is discussed. This part should be extended and clarified.

 Best Regards,

Author Response

Thank you for your kind review of our paper.

The expression of the purpose of this study in the introduction part has been carefully revised to show our clear scope. Please check if it works this time.